## METHOD

# CasKAS: direct profiling of genome-wide dCas9 and Cas9 specificity using ssDNA mapping

Georgi K. Marinov[1][*][†] , Samuel H. Kim[2][†], S. Tansu Bagdatli[1], Soon Il Higashino[1], Alexandro E. Trevino[3,4], Josh Tycko[1], Tong Wu[5], Lacramioara Bintu[4], Michael C. Bassik[1,6], Chuan He[5,7,8], Anshul Kundaje[1,9] and William J. Greenleaf[1,10,11,12*]

[†]Georgi K. Marinov and Samuel H. Kim contributed equally to this work.

*Correspondence:
GKM359@gmail.com;
wjg@stanford.edu

[1] Department of Genetics, School of Medicine, Stanford University, Stanford, CA 94305, USA
Full list of author information is available at the end of the article

## Abstract

Detecting and mitigating off-target activity is critical to the practical application of CRISPR-mediated genome and epigenome editing. While numerous methods have been developed to map Cas9 binding specificity genome-wide, they are generally time-consuming and/or expensive, and not applicable to catalytically dead CRISPR enzymes. We have developed CasKAS, a rapid, inexpensive, and facile assay for identifying off-target CRISPR enzyme binding and cleavage by chemically mapping the unwound single-stranded DNA structures formed upon binding of a sgRNA-loaded Cas9 protein. We demonstrate this method in both in vitro and in vivo contexts.

## Introduction

CRISPR-based methods for editing the genome and epigenome have emerged as a highly versatile means of manipulating the genetic makeup and regulatory states of cells. CRISPR technologies hold the potential to transform medical practice by enabling direct elimination of pathogenic sequence variants or manipulation of aberrant gene expression programs. CRISPR has also become a standard tool for discovery in biomedical research, including its uses for high-throughput, massively parallel genomic screens [1].

The presence of significant off-target effects is of universal concern for genome engineering technologies, presenting a major hurdle to fully realizing their potential utility. CRISPR tools have been shown to exhibit biochemical activity away from their intended target sites, which is particularly problematic for therapeutic applications, where risks of activity at sites other than the intended target leading to negative consequences to patient health must be minimal. Understanding and mapping these effects is therefore an urgent need.

To this end, numerous experimental approaches have been developed to experimentally map off-target effects genome-wide. Methods such as Digenome-seq [2] look for particular types of cut sites around target sequences in whole-genome sequencing data; however, deep whole-genome sequencing remains expensive. Assays such as BLESS [3], GUIDE-seq [4], HTGTS [5], DSBCapture [6], BLISS [7], SITE-seq [8], CIRCLE–seq [9], TTISS [10], INDUCE-seq [11], and CHANGE-seq [12] aim instead to directly map Cas9 cleavage events. However, all these methods involve some combination of complex and laborious molecular biology protocols and non-standard reagents and have not been widely adopted. Other methods, such as DISCOVER-seq [13], which maps DNA repair activity by applying ChIP-seq against the MRE11 protein, as well as earlier applications of ChIP-seq to map catalytically dead dCas9 occupancy sites genome-wide [14, 15], suffer from technical issues associated with the ChIP procedure. Most recently, long-read sequencing has been adapted to the problem of Cas9 specificity profiling, in the form of SMRT-OTS and Nano-OTS [16], but the cost of these methods is relatively high while their throughput is comparatively low.

These existing methods have differing advantages and weaknesses — some (e.g., ChIP-seq) are capable of capturing dCas9 association with DNA as a snapshot in time, others (e.g., those mapping editing outcomes by sequencing) provide information for off-target activity that can occur over a broad period of time, and generally with higher specificity.

Various computational models have also been trained to predict off-targets genome-wide [17, 18]. However, these exhibit far from perfect accuracy, and thus in many situations, especially within clinical contexts, direct experimental evidence is needed to accurately identify potential unintended effects of CRISPR-based reagents.

A faster, more accessible, and versatile method for mapping CRISPR off-targets is thus still a major need in the field. Here, we introduce CasKAS, a fast, inexpensive, and straightforward method for mapping CRPSIR off-targets that is applicable to both active and catalytically dead CRISPR enzymes. CasKAS takes advantage of the unwound single-strand DNA associated with CRISPR occupancy of DNA, which can be very specifically mapped using kethoxal as recently demonstrated by the KAS-seq assay. We demonstrate the application of CasKAS for profiling off-targets of active Cas9 and dCas9, in vitro using purified genomic DNA, and in vivo in live cells, and we also show that CasKAS can be used to distinguish off-target sites where active Cas9 cleaves DNA from sites where it is only binding. CasKAS is thus a highly versatile and adaptable tool for profiling CRISPR off-target sites, as well as for studying the dynamics of CRISPR association with the genome and of the editing process.

## Results

### CasKAS for mapping the physical association of CRISPR enzymes with DNA

When a Cas9-sgRNA ribonucleoprotein (RNP) is engaged with its target site, the sgRNA invades the DNA double helix, forming a ssDNA structure on the other strand (Fig. 1a). We thus reasoned that mapping ssDNA-containing regions should be a sensitive biochemical signal of productive Cas9 binding. The recently developed KAS-seq [19] assay for mapping single-stranded DNA (ssDNA) (*k*ethoxal-*a*ssisted **s**sDNA sequencing [19]) is ideally suited for the purpose of identifying ssDNA generated by CRISPR protein binding to DNA (Fig. 1a, b). KAS-seq is based on the specific covalent labeling of unpaired

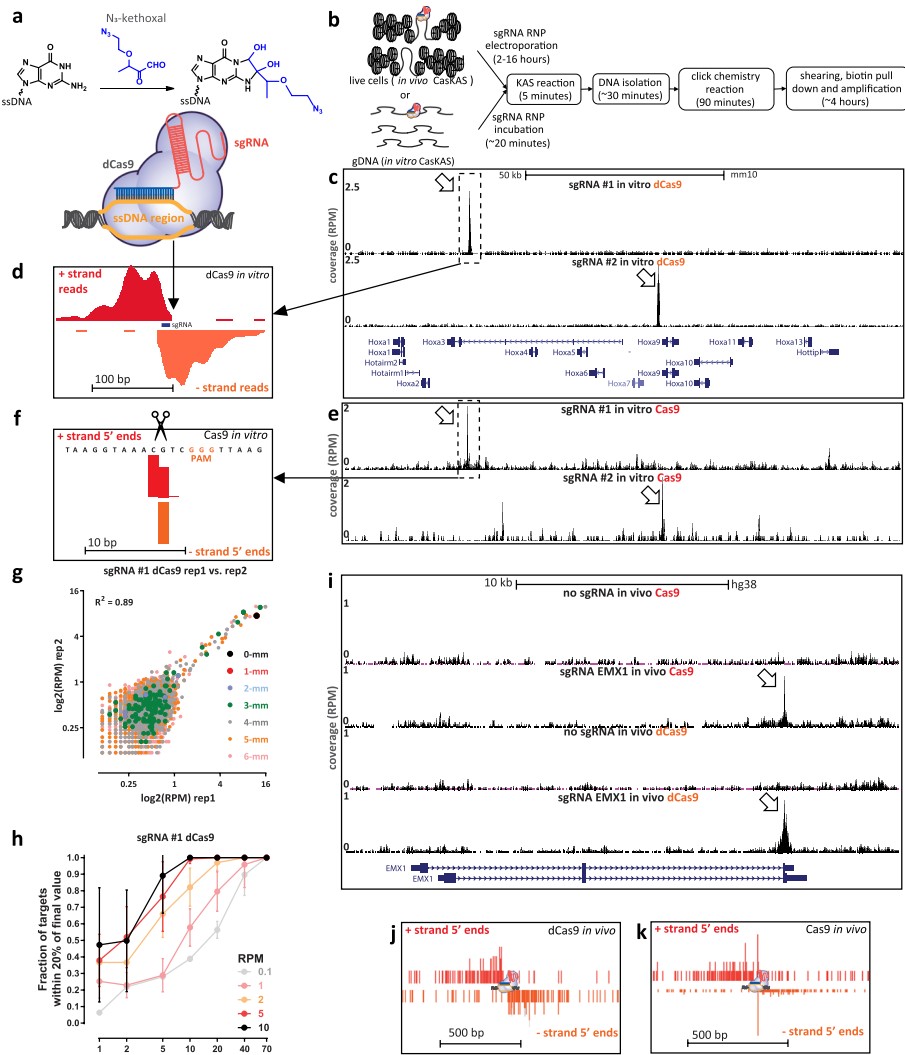

**Fig. 1** CasKAS maps dCas9- and Cas9-mediated strand invasion and cleavage events genome-wide in vitro on purified DNA and in vivo in cell lines. **a** CasKAS is based on the KAS-seq assay for mapping ssDNA structures. $N_3$-kethoxal covalently modifies unpaired guanine bases (while having no activity for G bases paired within dsDNA). Strand invasion by Cas9/dCas9 carrying an sgRNA results in the formation of a ssDNA structure, which can be directly identified using $N_3$-kethoxal. **b** Outline of in vivo and in vitro CasKAS. For in in vitro CasKAS, gDNA is incubated with a dCas9/Cas9 RNP, then $N_3$-kethoxal is added to the reaction; for in vivo CasKAS, cells are transfected with an RNP, then treated with kethoxal. DNA is then purified, click chemistry is carried out, DNA is sheared, labeled fragments are pulled down with streptavidin beads, and sequenced. **c**, **d** Mapping of dCas9 targets in vitro. **c** Mouse gDNA was incubated with dCas9 RNPs carrying one of two sgRNAs targeting the mouse *HOXA* locus. Highly specific labeling is observed at the expected target location of each sgRNA. **d** Asymmetric strand distribution of in vitro dCas9 CasKAS reads around the sgRNA target site. **e**, **f** Mapping of Cas9 targets in vitro. **e** Mouse gDNA was incubated with Cas9 RNPs carrying one of the same two sgRNAs targeting the mouse *HOXA* locus. **f** The distribution of 5′ read ends around targets sites in in vitro CasKAS datasets shows direct capture of the intermediate cleavage state. **g** Reproducibility of in vivo dCas9 CasKAS datasets. Shown are RPM values for 500 bp windows centered on the top ∼7000 predicted target sites for the "sgRNA #1" in two in vitro CasKAS experiments. Off-target sites are color-coded by the number of mismatches relative to the sgRNA. **h** CasKAS requires a moderate sequencing depth of $10–20 \times 10^6$ reads to accurately rank potential off-targets. A total of 10 different sets of subsamplings were generated, and the fraction of off-targets within 20% of their final quantification value was calculated for each. The mean $\pm$ SD is shown. **i**–**k** In vitro CasKAS maps Cas9 and dCas9 target sites. **i** Shown are CasKAS experiments with Cas9 and dCas9 and with the EMX1 sgRNA or with no sgRNA (negative control). **j** Assymmetric 5′ end distribution around target sites in dCas9 in vivo CasKAS. **k** In vivo Cas9 CasKAS, a mixture distribution is observed between phased cleavage sites and broader ssDNA labeling

guanine bases with $N_3$-kethoxal, generating an adduct to which biotin can then be added using click chemistry. After shearing, biotinylated DNA, corresponding to regions containing ssDNA structure, can be specifically enriched for and sequenced.

To determine the feasibility of using KAS-seq to map regions of ssDNA generated by Cas9 binding, we carried out an initial in vitro experiment using mouse genomic DNA (gDNA), purified dCas9 and two sgRNAs targeting the *Hoxa* locus.

Strikingly, we observed strong peaks at the expected target sites for each sgRNA (Fig. 1c). Detailed examination of dCas9 CasKAS profiles around the predicted sgRNA target sites revealed strand coverage asymmetry patterns similar to those observed for ChIP-seq around transcription factor binding sites [20] (Fig. 1d), indicating that enrichment derives from the sgRNA target site itself and confirming the utility of $N_3$-kethoxal for mapping dCas9 occupancy sites (in ChIP-seq, forward-strand reads are clustered to the left of the occupancy sites and reverse-strand reads to the right; this pattern arises because the occupancy site is crosslinked to the target protein and is thus always pulled down during immunoprecipitation resulting in all enriched fragments containing this site somewhere in their middle; observing a strong such pattern thus suggests high specificity of enrichment). We termed the assay "CasKAS."

### CasKAS for mapping active Cas9 nuclease cleavage sites

We then reasoned that CasKAS should also capture active Cas9 complexed with DNA, as the enzyme is thought to remain associated with DNA for some time after cleavage [21]. We performed CasKAS experiments with the same sgRNAs and active Cas9 nuclease, and again observed enrichment at the expected on-target sites (Fig. 1e). Examination of Cas9 CasKAS read profiles around the on-target site showed that the 5′ ends of reads are precisely positioned around the expected cut site, with one cut position on the target strand (which binds the sgRNA and is cleaved by the HNH domain) and two to three such positions on the non-target strand (which is cleaved by the RuvC domain; Fig. 1f), consistent with the previously known patterns of Cas9 cleavage [22, 23]. CasKAS therefore provides target specificity profiles for both active and catalytically dead Cas9 enzymes.

### CasKAS for mapping the activity of CRISPR enzymes in vivo

In vitro CasKAS data was highly reproducible between replicates (Fig. 1g), and a modest sequencing depth of between 10 and 20 million mapped reads was sufficient to capture off-target specificity profiles (Fig. 1h), which is an order of magnitude lower than required for resequencing the whole genome.

We observed similar results with two mouse sgRNAs targeting the *Nanog* locus (Additional file 1: Supplementary Fig. 1) and with two human sgRNA ("EMX1" and "VEGFA"; Additional file 1: Supplementary Fig. 2 and 3). We found no enrichment using components of the RNP in isolation — sgRNAs, dCas9 or Cas9 (Additional file 1: Supplementary Fig. 2).

Next, we tested the application of CasKAS in vivo in cell culture. Living cells contain substantial ssDNA due to active transcription, DNA replication, and other processes [19], so in vivo CasKAS signal derives from a mixture of Cas9associated ssDNA and endogenous processes. We carried out KAS-seq experiments using both dCas9 and Cas9

in HEK293 cells transfected with RNPs targeting *EMX1* or *VEGFA*, as well as negative, no-guide controls, which provided a map of background endogenous ssDNA profiles. At *EMX1*, which is not active in HEK293 cells, we observe strong peaks at the expected target site (Fig. 1i), as well as an asymmetric read profile around it for dCas9 (Fig. 1j), and a substantial degree of 5′ end clustering at the cut site, similar to what is observed in vitro for active Cas9 (Fig. 1g). The VEGFA gene is active in HEK293 cells, but the dCas9/Cas9 CasKAS signal is still readily identifiable as an addition to the endogenous ssDNA enrichment pattern (Additional file 1: Supplementary Fig. 4). These results demonstrate the utility of CasKAS for profiling CRISPR specificity both in vitro and in vivo.

We then examined the temporal dynamics of Cas9 and dCas9 association with the genome in vivo by carrying out in vivo time course for the EMX1 and VEGFA sgRNAs with both dCas9 and Cas9, assaying at 6, 12, 24, 48 and 72 h (Additional file 1: Supplementary Fig. 5–8). We find that association with DNA is not yet detectable at 6 h, is strongest at 48 h, and disappears for Cas9 at 72 h but persists for dCas9 at that time point. This is likely explained by the fact that by the 72-h time point cells have divided and DNA edits have been completed, thus disrupting Cas9's recognition of its cognate sequence. Thus, the 24-h and 48-h time points are optimal for in vivo CasKAS, with the caveat that this may be dependent on the growth dynamics of the cell lines/organisms being studied.

We further demonstrated the utility of CasKAS for profiling the association of Cas9 and dCas9 with the genome, both in vivo and in vitro, by carrying out CasKAS for pairs of guides targeting promoter regions of multiple human genes (*CD2*, CD90/*THY1*, CD45/*PTPRC*, CD298/*ATP1B3*) as well as a pair of "safe" sgRNAs targeting non-coding sequence. We observe similar patterns to those described above for the mouse #1 and #2, Nanog-sg2 and Nanog-sg3, and the human EMX1 and VEGFA sgRNAs (Additional file 1: Supplementary Fig. 9–16).

### Mapping CRISPR off-target sites using CasKAS

We next examined the genome-wide specificity of sgRNAs as measured by CasKAS. We focus on the mouse sgRNA #1 as it displayed a substantial number of off-targets yet that number was also sufficiently small for all of them to be examined directly. We first called peaks de novo (see the "Methods" section for details) without relying on off-target prediction algorithms, then manually curated the resulting peak set, excluding peaks not exhibiting the canonical asymmetric read distribution around a fixed point on the two strands (Fig. 2a; see also Additional file 1: Supplementary Fig. 17 for illustration). Remarkably, while we found 32 peaks at predicted off-target sites, we also found 198 (i.e., ∼6× as many) additional manually curated peaks; while these peaks exhibit generally lower CasKAS signal (Fig. 2b), they all display proper peak shape characteristics (see Additional file 1: Supplementary Fig. 17 for details), suggesting that they are genuine sites of occupancy. Most of the predicted (in total ∼7500) off-target sites for this sgRNA did not show substantial occupancy by dCas9 CasKAS (Fig. 2c, d).

Sequence comparison of the occupied predicted off-target sites allowed us to evaluate determinants of the specificity of dCas9 association and unwinding of DNA (Fig. 2e). Consistent with previous reports [24, 25], the PAM-distal region was much less sequence-constrained than the PAM-proximal seed region. We observed a similar

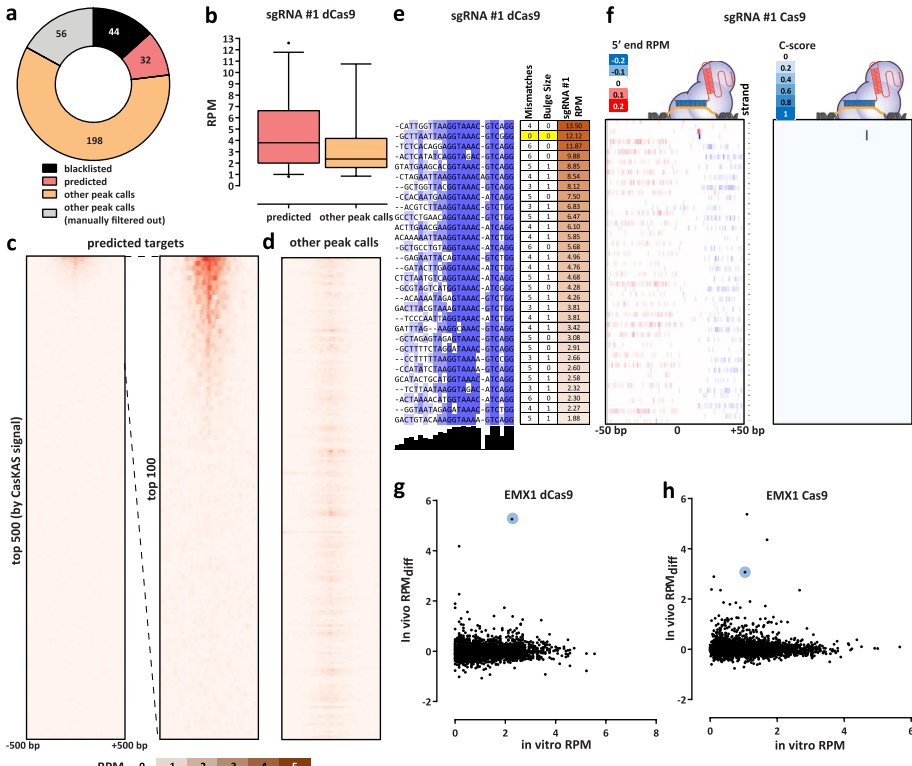

**Fig. 2** CasKAS profiles sgRNA specificity genome-wide. **a** Summary of de novo peak calls for sgRNA #1 (using MACS2). **b** CasKAS signal is stronger over predicted off-target sites, but legitimate interactions are also found elsewhere in the genome. **c** CasKAS profile over predicted (by Cas-OFFinder) off-target sites for sgRNA #1 with dCas9 (all such sites and focusing only on the top 100 ranked by dCas9 CasKAS signal). **d** CasKAS profile over peak calls outside predicted (by Cas-OFFinder) off-target sites for sgRNA #1 with dCas9. **e** Determinants of sequence specificity as measured by dCas9 CasKAS (for sgRNA #1). PAM-distal regions of the sgRNA are less constrained than its PAM-proximal parts. The on-target sgRNA is highlighted in yellow. **f** Active Cas9 signal read profiles can be used to distinguish off-targets associated with cutting from those where only binding occurs. Shown are the same off-target sites as in **e** and the plus- and minus-strand active Cas9 5′ end profiles around the sgRNA. In this case (sgRNA #1), only the on-target site shows a Cas9 CasKAS pattern indicating cleavage; at the other sites even active Cas9 likely only binds but does not cut. A simple cutting score metric ("*C*-score") based on multiplying the 5′ end forward- and reverse-strand profiles can be used to quantify cutting vs. binding. (g and h) Comparison between in vitro and in vivo CasKAS signal over predicted off-target sites for the EMX1 sgRNA. In vivo CasKAS is quantified as the difference in read per million (± 500 bp of the sgRNA site) between the sgRNA KAS-seq and the no-guide control KAS-seq ("RPM$_{diff}$"). The on-target site is shown in blue

pattern with the other sgRNAs we profiled, in both mouse and human (Additional file 1: Supplementary Fig. 30–41).

When analyzing peaks not associated with predicted off-target sites (Additional file 1: Supplementary Fig. 42) we observed other telling patterns — at numerous sites with strong dCas9 CasKAS signal, we observe a large number of mismatches to the sgRNA sequence as well as "bulge" regions wherein indels (relative to the sgRNA sequence) are observed in the target sequence. These mismatches and bulges were in general much larger than what is considered permissible by off-target prediction algorithms; we speculate that the lack of consideration of potential target sequences with large numbers of mismatches or substantial insertions could explain the much larger number of such sites compared to the set of occupied predicted off-targets.

We next devised a simple metric for evaluating the degree of read clustering at cut sites (a "*C*-score"; see Methods for details) to estimate the degree of cutting by Cas9. The on-target site exhibits the second highest dCas9 CasKAS signal genome-wide. However, strikingly, even though all CasKAS-identified off-target sites showed Cas9 binding, only the on-target site displayed strong cutting activity (Fig. 2f). The behavior of other sgR-NAs varies (Additional file 1: Supplementary Fig. 18–29 and 43–51), with some showing multiple clearly identifiable cut sites. Overall, these results are consistent with previous reports that Cas9 requires more successful RNA:DNA base pairing for cleavage activity than is necessary for binding [26, 27]. Thus, interpreting the read distributions of Cas9 CasKAS at target sites enables simultaneous detection of binding specificity and the promiscuity of catalytic activity.

We then carried out amplicon sequencing over a set of 81 potential EMX1 and 52 potential VEGFA on- and off-targets on genomic DNA extracted from HEK293 cells transfected with each of the two sgRNAs. These experiments generally corroborated the in vitro CasKAS results and identified no additional sites at which cutting occurs but for which cutting was not observed in in vitro CasKAS data (Additional file 1: Supplementary Fig. 52 and 53).

Finally, we compared in vitro and in vivo CasKAS profiles using the difference between the signal in CasKAS and no-guide negative control libraires as a measure of in vivo occupancy (Fig. 2g, h). We find many fewer strongly enriched sites in in vivo datasets than in vitro, with the on-target site being either the top (for dCas9) or among the top (for Cas9) sites in vivo. A potential explanation for this difference is the previously reported impediment of Cas9/dCas9 binding to DNA by the presence of nucleosomes [28].

## Discussion

In conclusion, we have presented CasKAS, a simple and robust method for mapping the specificity of active and catalytically dead versions of CRISPR enzymes. CasKAS has numerous advantages over existing tools while also opening up new possibilities for studying CRISPR biology. CasKAS requires no specialized molecular biology protocols, takes just a few hours in vitro (and a similar amount of time after harvesting cells in vivo), and, due to the strong, active enrichment of target sequences, is inexpensive. In contrast to previously developed methods, It measures strand invasion by CRISPR, which is likely biochemically more specific and relevant to CRISPR function than DNA association (invasion is critical for cleavage by active Cas9, and it also ensures stable occupancy to drive epigenetic modulation and the various other effector functions of dCas9 fusions). We compared de novo called CasKAS peaks to those generated by other means, and while we found a variable degree of concordance and large sets of peaks unique to some methods, those found only by CasKAS often contained higher fractions of predicted off-target sites than those unique to other methods (Additional file 1: Supplementary Fig. 54–58).

CasKAS does not rely on measuring DNA cleavage or modification and can thus be used to profile the specificity of all types of DNA-targeting CRISPR proteins that generate a stable ssDNA structure. CasKAS also does not rely on cellular repair processes, cell division, or delivery of additional exogenous DNA (as in GUIDE-seq) to generate a detectable

signal. These advantages, coupled with low cell input requirements, may increase the utility of the method in rare primary cell types, tissues from animal models, or even for direct assessment of specificity in edited patient cells (e.g., ex vivo edited immune cells). A current limitation of CasKAS is the requirement that a G nucleotide is present within the sgRNA sequence, since kethoxal requires an exposed G to react with. However, only a small fraction ($\leq 5\%$) of sgRNAs in the human genome lack any Gs for *S. pyogenes* PAM sequences (Additional file 1: Supplementary Fig. 59). We also do not observe a strong correlation between the number of G bases in an off-target sgRNA match and CasKAS enrichment (Additional file 1: Supplementary Fig. 60). A minor limitation specific to in vivo experiments is that high levels of ssDNA generated as a result of active transcription or other endogenous processes may obscure the CasKAS signal at certain loci in some situations. We have explored this issue with sgRNAs targeting the promoter of the CD298/*ATP1B3* gene (Additional file 1: Supplementary Fig. 16), where we observe additional KAS signal well above the endogenous levels for dCas9 but not for the active Cas9; this suggests that dCas9, the association of which does not result in cuts to DNA, is likely able to reassociate with DNA if displaced by transcription (or other processes); in contrast, active Cas9 is not. Another minor limitation of the current in vitro protocol is that labeling is carried out on high molecular weight (HMW) DNA and samples must be sheared serially. We have explored using pre-sheared and end-repaired DNA (to minimize kethoxal labeling of Gs on sticky ends generated by sonication), with comparable results to using HMW DNA (Additional file 1: Supplementary Fig. 61); we anticipate that further optimization or using other approaches, such as enzymatic fragmentation, should allow the parallel high-throughput plate-based profiling of the specificity of very large numbers of sgRNAs.

In addition to being highly valuable for off-target profiling in vitro and in previously difficult to assay settings such as primary cells, we expect CasKAS to provide fruitful insights into the mechanisms and dynamics of in vivo CRISPR action (taking advantage of finely controllable CRISPR systems such as vfCRISPR [29]), and the influence of transcriptional, regulatory, and epigenetic and other functional genomic contexts on CRISPR activity.

## Conclusions

We have presented CasKAS, a new and highly versatile method for profiling CRISPR protein activity genome-wide that takes advantage of the formation of single-stranded DNA structures during the association of CRISPR proteins with DNA. CasKAS can be used both in vitro and in vivo, and with both catalytically active and inactive versions of the CRISPR enzymes. When catalytically active CRISPR proteins are used, CasKAS not only maps the general location of on and off-targets, but also the precise positions of cleavage events. We expect CasKAS to become a highly useful, easily accessible tool for both mapping CRISPR off-targets and for tracking CRISPR activity in vivo in a wide variety of contexts.

## Methods

### Guide RNA sequences

Guide RNAs were obtained from IDT ("sgRNA #1" and "sgRNA #2") or from Synthego (all others). The following sgRNA sequences were used in this study:

1. "sgRNA #1": GCTTAATTAAGGTAAACGTC

2. "sgRNA #2": CCAACCTGGCGGCTCGTTGG
3. "EMX1 Tsai": GAGTCCGAGCAGAAGAAGAA
4. "VEGFA-site1": GGGTGGGGGGGAGTTTGCTCC
5. "Nanog-sg2": GATCTCTAGTGGGAAGTTTC
6. "Nanog-sg3": GTCTGTAGAAAGAATGGAAG
7. "CD2-1": ACATGGAAAGCTCATCTTAG
8. "CD2-2": TACATGGAAAGCTCATCTTA
9. "CD90-1": GCGGAAGACCCCAGTCCAGG
10. "CD90-2": GTCCAGGTGGGAACTGGAGC
11. "CD45-1": GTTTGTTCTTAGGGTAACAG
12. "CD45-2": GAGTTTAAGCCACAAATACA
13. "CD298-1": GACGGCAGTGAAGGGTGGGA
14. "CD298-2": GAGTACTCCCCGTAACGAGG
15. "safe-1": GTGCATTGTTGGTGGTTGTG
16. "safe-2'": GCTAAAGTATCAAAGGGAAT

Guide RNAs were dissolved to a concentration of 100 μM using nuclease-free $1 \times$ TE buffer and stored at $-20$ °C.

## In vitro CasKAS
In vitro CasKAS experiments were executed as follows.

First, 1 μL of each synthetic sgRNA was incubated at room temperature with 1 μL of recombinant purified dCas9 (MCLab dCAS9B-200, at 20 μM, i.e., a total of 20 pmol) for 20 min. The RNP was then incubated with 1 μg of gDNA at 37 °C for 10 min.

The KAS reaction was then carried out by adding 1 μL of 500 mM $N_3$-kethoxal (ApeXBio A8793). DNA was immediately purified using the MinElute PCR Purification Kit (Qiagen 28,006) and eluted in 87.5 or 175 μL 25 mM $K_3BO_3$.

## In vivo CasKAS
For in vivo CasKAS experiments, HEK293T cells were seeded at 400,000 cells/well into a 6-well plate the day before RNP transfection. Media was exchanged 2 h before transfection. For each well, 6250 ng of Cas9 (MCLAB CAS9-200) or dCas9 (MCLAB dCAS9B-200) and 1200 ng sgRNA was complexed with CRISPRMAX (Thermo Fisher CMAX00008) reagent in Opti-MEM (Thermo Fisher 51,985,091) following manufacturer's protocol. After incubation at room temperature for 15 min, the RNP solution was directly added to each well and gently mixed. The cells were incubated with the RNP complex for 14 h at 37 °C. To harvest and perform kethoxal labeling, media was removed and room temperature $1 \times$ PBS was used to wash the cells. Cells were then dissociated with trypsin, trypsin was quenched with media, cells were pelleted at room temperature, and then resuspended in 100 μL of media supplemented with 5 mM $N_3$-kethoxal (final concentration). Cells were incubated for 10 min at 37 °C with shaking at 500 rpm in a Thermomixer. Cells were then pelleted by centrifuging at 500 g for 5 min at 4 °C. Genomic DNA was then extracted using the Monarch gDNA Purification Kit (NEB T3010S) following the standard protocol but with elution using 175 μL 25 mM $K_3BO_3$ at pH 7.0.

**Click reaction, biotin pull down and library generation**

The click reaction was carried out by combining 175 μL purified DNA, 5 μL 20 mM DBCO-PEG4-biotin (DMSO solution, Sigma 760,749), and 20 μL 10 × PBS in a final volume of 200 μL or 87.5 μL purified and sheared DNA, 2.5 μL 20 mM DBCO-PEG4-biotin (DMSO solution, Sigma 760,749), and 10 μL 10 × PBS in a final volume of 100 μL. The reaction was incubated at 37 °C for 90 min.

DNA was purified using AMPure XP beads (50 μL for a-100 μL reaction or 100 μL for a-200 μL reaction); beads were washed on a magnetic stand twice with 80% EtOH and eluted in 130 μL 25 mM $K_3BO_3$.

Purified DNA was then sheared on a Covaris E220 instrument down to ∼150–400 bp size.

For streptavidin pulldown of biotin-labeled DNA, 10 μL of 10 mg/mL Dynabeads MyOne Streptavidin T1 beads (Life Technologies, 65,602) were separated on a magnetic stand, then washed with 300 μL of 1 × TWB (Tween Washing Buffer; 5 mM Tris–HCl pH 7.5; 0.5 mM EDTA; 1 M NaCl; 0.05% Tween 20). The beads were resuspended in 300 μL of 2 × Binding Buffer (10 mM Tris–HCl pH 7.5, 1 mM EDTA; 2 M NaCl), the sonicated DNA was added (diluted to a final volume of 300 μL if necessary), and the beads were incubated for ≥ 15 min at room temperature on a rotator. After separation on a magnetic stand, the beads were washed with 300 μL of 1 × TWB, and heated at 55 °C in a Thermomixer with shaking for 2 min. After removal of the supernatant on a magnetic stand, the TWB wash and 55 °C incubation were repeated.

Final libraries were prepared on beads using the NEBNext Ultra II DNA Library Prep Kit (NEB, #E7645) as follows. End repair was carried out by resuspending beads in 50 μL 1 × EB buffer, and adding 3 μL NEB Ultra End Repair Enzyme and 7 μL NEB Ultra End Repair Enzyme, followed by incubation at 20 °C for 30 min (in a Thermomixer, with shaking at 1000 rpm) and then at 65 °C for 30 min.

Adapters were ligated to DNA fragments by adding 30 μL Blunt Ligation mix, 1 μL Ligation Enhancer and 2.5 μL NEB Adapter, incubating at 20 °C for 20 min, adding 3 μL USER enzyme, and incubating at 37 °C for 15 min (in a Thermomixer, with shaking at 1000 rpm).

Beads were then separated on a magnetic stand, and washed with 300 μL TWB for 2 min at 55 °C, 1000 rpm in a Thermomixer. After separation on a magnetic stand, beads were washed in 100 μL 0.1 × TE buffer, then resuspended in 15 μL 0.1 × TE buffer, and heated at 98 °C for 10 min.

For PCR, 5 μL of each of the i5 and i7 NEB Next sequencing adapters were added together with 25 μL 2 × NEB Ultra PCR Mater Mix. PCR was carried out with a 98 °C incubation for 30 s and 12 cycles of 98 °C for 10 s, 65 °C for 30 s, and 72 °C for 1 min, followed by incubation at 72 °C for 5 min.

Beads were separated on a magnetic stand, and the supernatant was cleaned up using 1.8 × AMPure XP beads.

Libraries were sequenced in a paired-end format on an Illumina NextSeq instrument using NextSeq 500/550 high output kits (2 × 36 cycles).

## CasKAS data processing

Demultipexed fastq files were mapped to the hg38 assembly of the human genome or the mm10 version of the mouse genome as $2 \times 36$mers using Bowtie [30] with the following settings: -v 2 -k 2 -m 1 –best –strata -X 1000. Duplicate reads were removed using picard-tools (version 1.99).

Browser tracks generation, fragment length estimation, TSS enrichment calculations, and other analyses were carried out as previously described [31, 32] using custom-written Python scripts (https://github.com/georgimarinov/GeorgiScripts). The refSeq set of annotations were used for evaluation of enrichment around TSSs.

## CasKAS peak calling

Peak calling on in vitro binding datasets was carried out using version 2.1.0 of MACS2 [33] with default settings. Peaks were then compared against the ENCODE set of "blacklisted" regions [34] to filter out likely artifacts.

## Sequence analysis

Guide RNA off-target predictions were obtained from Cas-OFFinder [35].

Multiple sequence alignments of sgRNA sequences and their off-targets were generated using MUSCLE [36] and visualized using JalView [37].

## Quantification

### CasKAS occupancy quantification

For in vitro CasKAS datasets, we quantified occupancy by calculating Read-Per-Million (RPM) values for the $\pm 500$-bp regions around off-target sites using the traditional RPM formula:

$$RPM_{OT} = \frac{|R_{OT}|}{\frac{|R|}{10^6}} \tag{1}$$

where $|R_{OT}|$ is the number of reads mapping to the $\pm 500$-bp off-target region, and $|R|$ is the total number of mapped reads.

For in vitro CasKAS datasets, we estimated occupancy levels as the difference between in vivo CasKAS RPM values and RPM values in a negative no-sgRNA control.

## Cutting score calculation

The Cas9 cutting *C*-score was calculated as follows.

First, basepair-level RPM profiles for mapped read 5′ ends were generated separately for the forward and reverse strands. Then the *C*-score was calculated by multiplying the forward and reverse strand profiles (summed over a running window of 3 bp):

$$C-\text{score}_{c,i} = \sum_{j=i-1}^{j=i+1} RPM_{c,j}^{+} \times \sum_{j+i-1}^{j=i+1} RPM_{c,j}^{-} \tag{2}$$

where *c,i* indicate the coordinates by chromosome and position.

### Amplicon sequencing

Amplicon sequencing was performed according to the xGen Amplicon Panels (IDT) protocol. A custom panel of primers was designed for predicted off-target sites for each sgRNA through IDT's amplicon sequencing panel design service. Multiplex amplicon PCR was performed using 17.5 ng of genomic DNA for PCR. For amplification, the standard xGen Amplicon Panels protocol was followed with annealing at 63 °C and extension at 65 °C. Libraries were quantified using NEBNext Library Quantification Kit for Illumina (E7630S) and pooled for sequencing on a MiSeq.

### Amplicon sequencing analysis

Amplicon sequencing reads were aligned against the hg38 version of the human genome using bwa mem [38] (version 0.7.5a) with default settings. Indel frequencies per basepair were calculated as the fraction of reads containing an indel over a given position using custom-written scripts.

## Supplementary Information

---

**Additional file 1: Supplementary Figure 1.** In vitro dCas9 and Cas9 CasKAS profiles around the mouse Nanog locus using the "Nanog-sg2" and "Nanog-sg3" sgRNAs. **Supplementary Figure 2.** CasKAS signal in vitro is specific to the activity of the dCas9/Cas9 protein combined with its sgRNA. **Supplementary Figure 3.** CasKAS signal in vitro around the VEGFA gene with the VEGFA sgRNA. **Supplementary Figure 4.** CasKAS signal in vivo around the VEGFA gene with the VEGFA sgRNA. **Supplementary Figure 5.** Time course of in vivo CasKAS signal around the EMX1 gene with the EMX1 sgRNA using dCas9. **Supplementary Figure 6.** Time course of in vivo CasKAS signal around the VEGFA gene with the VEGFA sgRNA using dCas9. **Supplementary Figure 7.** Time course of in vivo CasKAS signal around the EMX1 gene with the EMX1 sgRNA using active Cas9. **Supplementary Figure 8.** Time course of in vivo CasKAS signal around the VEGFA gene with the VEGFA sgRNA using active Cas9. **Supplementary Figure 9.** CasKAS signal in vitro around the CD2 gene with two different sgRNA targeting the gene. **Supplementary Figure 10.** CasKAS signal in vivo (HEK293 cells, harvested at 48 hours) around the CD2 gene with two different sgRNA targeting the gene. **Supplementary Figure 11.** CasKAS signal in vitro around the CD90/THY1 gene with two different sgRNA targeting the gene. **Supplementary Figure 12.** CasKAS signal in vivo (HEK293 cells, harvested at 48 hours) around the CD90/THY1 gene with two different sgRNA targeting the gene. **Supplementary Figure 13.** CasKAS signal in vitro around the CD45/PTPRC gene with two different sgRNA targeting the gene. **Supplementary Figure 14.** CasKAS signal in vivo (HEK293 cells, harvested at 48 hours) around the CD45/PTPRC gene with two different sgRNA targeting the gene. **Supplementary Figure 15.** CasKAS signal in vitro around the CD298/ATP1B3 gene with two different sgRNA targeting the gene. **Supplementary Figure 16.** CasKAS signal in vivo (HEK293 cells, harvested at 48 hours) around the CD298/ATP1B3 gene with two different sgRNA targeting the gene. **Supplementary Figure 17.** CasKAS identifies proper off-target sites that are missed by sgRNA prediction algorithms. **Supplementary Figure 18.** In vitro dCas9 and Cas9 CasKAS profiles for the "Nanog-sg2" sgRNA. **Supplementary Figure 19.** In vitro dCas9 and Cas9 CasKAS profiles for the "Nanog-sg3" sgRNA. **Supplementary Figure 20.** In vitro dCas9 and Cas9 CasKAS profiles for the "EMX1 Tsai" sgRNA. **Supplementary Figure 21.** In vitro dCas9 and Cas9 CasKAS profiles for the "VEGFA-site1" sgRNA. **Supplementary Figure 22.** In vitro dCas9 and Cas9 CasKAS profiles for the "CD2-1" sgRNA. **Supplementary Figure 22.** In vitro dCas9 and Cas9 CasKAS profiles for the "CD2-1" sgRNA. **Supplementary Figure 23.** In vitro dCas9 and Cas9 CasKAS profiles for the "CD2-2" sgRNA. **Supplementary Figure 24.** In vitro dCas9 and Cas9 CasKAS profiles for the "CD45-1" sgRNA. **Supplementary Figure 25.** In vitro dCas9 and Cas9 CasKAS profiles for the "CD45-2" sgRNA. **Supplementary Figure 26.** In vitro Cas9 CasKAS profiles for the "CD90-1" sgRNA. **Supplementary Figure 27.** In vitro dCas9 and Cas9 CasKAS profiles for the "CD90-2" sgRNA. **Supplementary Figure 28.** In vitro dCas9 and Cas9 CasKAS profiles for the "CD298-1" sgRNA. **Supplementary Figure 29.** In vitro dCas9 and Cas9 CasKAS profiles for the "CD298-2" sgRNA. **Supplementary Figure 30.** Multiple sequence alignment of offtarget sites identified by in vitro dCas9 and Cas9 CasKAS for the "Nanog-sg2" sgRNA. **Supplementary Figure 31.** Multiple sequence alignment of offtarget sites identified by in vitro dCas9 and Cas9 CasKAS for the "Nanog-sg3" sgRNA. **Supplementary Figure 32.** Multiple sequence alignment of offtarget sites identified by in vitro dCas9 and Cas9 CasKAS for the "EMX1 Tsai" sgRNA. **Supplementary Figure 33.** Multiple sequence alignment of offtarget sites identified by in vitro dCas9 and Cas9 CasKAS for the "VEGFA-site1" sgRNA. **Supplementary Figure 34.** Multiple sequence alignment of offtarget sites identified by in vitro dCas9 and Cas9 CasKAS for the "CD2-1" sgRNA. **Supplementary Figure 35.** Multiple sequence alignment of offtarget sites identified by in vitro dCas9 and Cas9 CasKAS for the "CD-2" sgRNA. **Supplementary Figure 36.** Multiple sequence alignment of offtarget sites identified by in vitro dCas9 and Cas9 CasKAS for the "CD45-1" sgRNA. **Supplementary Figure 37.** Multiple sequence alignment of off-target sites identified by in vitro dCas9 and

Cas9 CasKAS for the "CD45-2" sgRNA. **Supplementary Figure 38.** Multiple sequence alignment of off-target sites identified by in vitro Cas9 CasKAS for the "CD90-1" sgRNA. **Supplementary Figure 39.** Multiple sequence alignment of offtarget sites identified by in vitro dCas9 and Cas9 CasKAS for the "CD90-2" sgRNA.. **Supplementary Figure 40.** Multiple sequence alignment of off-target sites identified by in vitro dCas9 and Cas9 CasKAS for the "CD298-1" sgRNA. **Supplementary Figure 41.** Multiple sequence alignment of off-target sites identified by in vitro dCas9 and Cas9 CasKAS for the "CD298-2" sgRNA. **Supplementary Figure 42.** Multiple sequence alignment of off-target sites identified by in vitro dCas9 and Cas9 CasKAS for the "sgRNA #1" sgRNA outside the list of predicted off-targets by Cass-OFFinder. **Supplementary Figure 43.** Cutting profiles around on- and off-target sites for the VEGFA sgRNA. **Supplementary Figure 44.** Cutting profiles around the top 100 on- and off-target sites for the "CD2-1" sgRNA. **Supplementary Figure 45.** Cutting profiles around the top 100 on- and off-target sites for the "CD2-2" sgRNA. **Supplementary Figure 46.** Cutting profiles around the top 100 on- and off-target sites for the "CD45-1" sgRNA. **Supplementary Figure 47.** Cutting profiles around the top 100 on- and off-target sites for the "CD45-2" sgRNA. **Supplementary Figure 48.** Cutting profiles around the top 100 on- and off-target sites for the "CD90-1" sgRNA. **Supplementary Figure 49.** Cutting profiles around the top 100 on- and off-target sites for the "CD90-2" sgRNA. **Supplementary Figure 50.** Cutting profiles around the top 100 on- and off-target sites for the "CD298-1" sgRNA. **Supplementary Figure 51.** Cutting profiles around the top 100 on- and off-target sites for the "CD298-2" sgRNA. **Supplementary Figure 52.** Amplicon sequencing of DNA edits with the EMX1 sgRNA. **Supplementary Figure 53.** Amplicon sequencing of DNA edits with the VEGFA sgRNA. **Supplementary Figure 54.** Comparing in vitro dCas9 results to using ChIP-seq and CHANGE-seq for off-target profiling. **Supplementary Figure 55.** Comparing in vitro dCas9 results to using DISCOVER-seq for off-target profiling. **Supplementary Figure 56.** Comparing in vitro dCas9 results to using DISCOVER-seq for off-target profiling. **Supplementary Figure 57.** Comparing in vitro dCas9 results to using GUIDE-seq for off-target profiling. **Supplementary Figure 58.** Comparing in vitro dCas9 results to using Digenome-seq for off-target profiling. **Supplementary Figure 59.** Most sgRNAs in the human genome contain multiple G nucleotides and are thus subject to labeling by N3-kethoxal. **Supplementary Figure 60.** Absence of strong correlation between the number of G nucleotides in a sgRNA off-target site and CasKAS signal. **Supplementary Figure 61.** CasKAS can be performed on pre-sheared DNA.

**Additional file 2.** Review history.

## Acknowledgements
The authors would like to thank Zohar Shipony and members of the Greenleaf, Kundaje, and Bassik labs for helpful discussion and suggestions regarding this work.

## Review history
The review history is available as Additional file 2.

## Peer review information

## Authors' contributions
G.K.M. conceptualized the study, performed initial in vitro CasKAS experiments, analyzed data, and wrote the manuscript with input from all authors. S.H.K. developed the in vivo CasKAS protocol and performed in vivo CasKAS experiments together with S.I.H. S.T.B. carried out in vitro CasKAS optimization. A.E.T. and J.T. supplied sgRNAs and designed off-target profiling experiments. A.E.T. carried out off-target analysis for mouse sgRNAs. T.W. provided key reagents. W.J.G., A.K., C.H. M.C.B., and L.B. supervised the study. The author(s) read and approved the final manuscript.

## Funding
This work was supported by NIH grants (P50HG007735, RO1 HG008140, U19AI057266 and UM1HG009442 to W.J.G., 1UM1HG009436 to W.J.G. and A.K., 1DP2OD022870-01 and 1U01HG009431 to A.K., and HG006827 to C.H.), the Rita Allen Foundation (to W.J.G.), the Baxter Foundation Faculty Scholar Grant, and the Human Frontiers Science Program grant RGY006S (to W.J.G). W.J.G is a Chan Zuckerberg Biohub investigator and acknowledges grants 2017–174468 and 2018–182817 from the Chan Zuckerberg Initiative. S.K. is supported by MSTP training grant T32GM007365 and the Paul and Daisy Soros Fellowship. J.T. is supported by the NIDDK F99/K00 fellowship of the National Institutes of Health (F99DK126120). M.C.B. is supported by a grant from Stanford ChEM-H and an NIH Director's New Innovator Award (1DP2HD08406901). Fellowship support also provided by the Stanford School of Medicine Dean's Fellowship (G.K.M.), the Siebel Scholars, the Enhancing Diversity in Graduate Education Program and the Weiland Family Fellowship (A.E.T.). C.H. is a Howard Hughes Medical Institute Investigator.

## Availability of data and materials
Sequencing reads for the datasets described in this study are available from GEO accession GSE171962 [39].

# Declarations

## Ethics approval and consent to participate
Not applicable for this study.

## Competing interests
G.K.M., W.J.G, T.W., and C.H. have submitted a provisional patent application based on this work. This patent will not limit the non-commercial use of CasKAS or the ability to reproduce the results of the study.

## Author details
[1]Department of Genetics, School of Medicine, Stanford University, Stanford, CA 94305, USA. [2]Cancer Biology Program, School of Medicine, Stanford University, Stanford, CA 94305, USA. [3]Center for Personal Dynamic Regulomes, Stanford University, Stanford, CA 94305, USA. [4]Department of Bioengineering, Stanford University, Stanford, CA 94305, USA. [5]Department of Chemistry, The University of Chicago, Chicago, IL 60637, USA. [6]Chemistry, Engineering, and Medicine for Human Health (ChEM-H), Stanford University, Stanford, CA 94305, USA. [7]Department of Biochemistry and Molecular Biology and Institute for Biophysical Dynamics, The University of Chicago, Chicago, IL 60637, USA. [8]Howard Hughes Medical Institute, The University of Chicago, Chicago, IL 60637, USA. [9]Department of Computer Science, Stanford University, Stanford, CA 94305, USA. [10]Department of Applied Physics, Stanford University, Stanford, CA 94305, USA. [11]Center for Personal Dynamic Regulomes, Stanford University, Stanford, CA 94305, USA. [12]Chan Zuckerberg Biohub, San Francisco, CA, USA.

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

## 

