## [**Additional file 2.** Review history. · Genome Biology]

Review History

First round of review

Reviewer 1

Are you able to assess all statistics in the manuscript, including the appropriateness of statistical tests used? Yes, and I have assessed the statistics in my report.

Comments to author:

Summary

In this study, the authors presented a method for conveniently and accurately detecting target specificity for a specific guide for the CRISPR-Cas9 system. The authors targeted the ssDNA exposure caused by CRISPR-Cas9 binding and detected off-target corresponding to off-target binding and cleavage using click chemistry.

This study is very interesting, and it is considered that the ripple effect will be very large in the future by conveniently and accurately confirming off-target measurement in related fields. I recommend that this paper be published in the journal 'Genome Biology'.

Minor

Regarding Figures 1i,

Although the authors mentioned background signal processing on page 4, lanes 48-54, there is a lack of precise description of how to exclude background signals such as transcription when measuring off-target in target cells. For example, whether the background signal was excluded at the same time-point during CRISPR-Cas9 treated and non-treated, etc., an accurate explanation is required.

In page 6, lane 3-8,

'CasKAS does not rely on measuring DNA cleavage or modification and can thus be used to profile the specificity

of all types of DNA-targeting CRISPR proteins' This sentence doesn't seem appropriate.

Depending on the binding of the CRISPR-Cas type, it is thought to be applicable to the type in which ssDNA is exposed to the solution. In addition, for universal application of CasKAS technology, it is recommended to suggest that it can be applied to the latest technology such as prime editing based on the Cas9 system where ssDNA is not exposed. If prime editing off-target candidates are preliminary selected for a specific guide RNA in cells using Cas9, it is thought that additional validation will be possible using targeted amplicon seq.

Reviewer 2

Are you able to assess all statistics in the manuscript, including the appropriateness of statistical tests used? Yes, and I have assessed the statistics in my report.

Comments to author:

Marinov et al propose a novel method (casKas) to measure target and off-target activity of cas9. This methodology differs from most alternatives as it is compatible with dead versions of cas9 (dcas9). Cas9 targeting is measured using N3-kethoxal-assisted labeling (Wu et al, Nat Methods 2020) which allows labeling of guanines in ssDNA. This method also differs from ChiSeq based approaches as it is more straightforward and with different biases. Also, casKAS seems to be advantageous as it very likely works with any CRISPR and ancestral system (all described to date expose ssDNA) and it seems a very interesting tool to study mechanistic aspects of cas9/dcas9.

Major comments

#1

As the method is targeting guanines it would be important to show more accurately the sensitivity to guanosine presence in interrogated site. I am not totally convinced by Supp. Fig. 60 which displays overall correlation. It would help to provide specific analysis or measurements within the area with a signal above noise.

#2

This sentence should be rephrased: "It measures strand invasion by CRISPR, which is biochemically more specific and relevant to CRISPR function than DNA association." It is unclear that this statement is universal. For instance, certain applications of CRISPR do not require helix opening (ie Epigenetic modulation)

#3

DNA replication generates transient ssDNA exposure, also other regions such as centromeres and telomeres expose ssDNA. It would be positive to comment if this interferes with the measurement.

Minor comments

#1 text in both figures should be made bigger. It results very hard to read the text.

Direct profiling of genome-wide dCas9 and Cas9 specificity using ssDNA mapping (CasKAS).

Response to reviewer comments

Reviewer 1:

Summary

In this study, the authors presented a method for conveniently and accurately detecting target specificity for a specific guide for the CRISPR-Cas9 system. The authors targeted the ssDNA exposure caused by CRISPR-Cas9 binding and detected off-target corresponding to off-target binding and cleavage using click chemistry.

This study is very interesting, and it is considered that the ripple effect will be very large in the future by conveniently and accurately confirming off-target measurement in related fields. I recommend that this paper be published in the journal 'Genome Biology'.

We thank the reviewer for the comments and suggestions.

Minor

Regarding Figures 1i,

Although the authors mentioned background signal processing on page 4, lanes 48-54, there is a lack of precise description of how to exclude background signals such as transcription when measuring off-target in target cells. For example, whether the background signal was excluded at the same time-point during CRISPR-Cas9 treated and non-treated, etc., an accurate explanation is required.

For Figure 1i specifically, no background subtraction was carried out. The purpose of Figure 1i is to illustrate the raw properties of the data. For quantification of *in vivo* CasKAS signal at putative off-target sites, we subtract the KAS-seq signal from matched negative no-sgRNA controls from the *in vivo* CasKAS signal, as shown in Figure 2g-h. We have added a further clarification on that in the main text and a new subsection in the Methods section.

In page 6, lane 3-8, 'CasKAS does not rely on measuring DNA cleavage or modification and can thus be used to profile the specificity of all types of DNA-targeting CRISPR proteins' This sentence doesn't seem appropriate.

Depending on the binding of the CRISPR-Cas type, it is thought to be applicable to the type in which ssDNA is exposed to the solution. In addition, for universal application of CasKAS technology, it is recommended to suggest that it can be applied to the latest technology such as prime editing based on the Cas9 system where ssDNA is not exposed. If prime editing off-target candidates are preliminary selected for a specific guide RNA in cells using Cas9, it is thought that additional validation will be possible using targeted amplicon seq.

We have revised this sentence with the caveat that a ssDNA structure needs to be generated.

We have not yet tested CasKAS in the prime editing context, but it is far from certain that ssDNA is not exposed at all in the process – it certainly is during the initial nicking, and then at the subsequent steps not all DNA is fully paired at the edges of the bubble. One potential application of CasKAS is in fact for figuring out the temporal dynamics of the structure of these intermediates (e.g. see PMID: 32732424).

Reviewer 2:

Marinov et al propose a novel method (casKas) to measure target and off-target activity of cas9. This methodology differs from most alternatives as it is compatible with dead versions of cas9 (dcas9). Cas9 targeting is measured using N3-kethoxal-assisted labeling (Wu et al, Nat Methods 2020) which allows labeling of guanines in ssDNA. This method also differs from ChiSeq based approaches as it is more

straightforward and with different biases. Also, casKAS seems to be advantageous as it very likely works with any CRISPR and ancestral system (all described to date expose ssDNA) and it seems a very interesting tool to study mechanistic aspects of cas9/dcas9.

We thank the reviewer for the comments and suggestions.

Major comments

#1

As the method is targeting guanines it would be important to show more accurately the sensitivity to guanosine presence in interrogated site. I am not totally convinced by Supp. Fig. 60 which displays overall correlation. It would help to provide specific analysis or measurements within the area with a signal above noise.

We agree with the reviewer that the requirement for guanines is an important consideration, and it is a concern we ourselves have had, addressed previously, and further elaborated on in the revised manuscript.

Unfortunately, there is no experimental way that we are currently aware of to directly address this issue as KAS-seq is the only available method for measuring ssDNA with high resolution and signal-to-noise ratio, thus there is no orthogonal measure that we can use to assess the relationship between the presence and number of Gs at off-target sites and the representation of those sites in final sequencing libraries.

Association of Cas9 with DNA can be measured by *in vivo* or *in vitro* ChIP-seq, but a similar limitation applies there too, as ChIP is dependent on cross-linking, and cross-linking happens most efficiently between guanines on DNA and lysines and arginines on proteins. It is therefore also confounded to a certain extent by base composition (this was in fact one of the original motivations behind developing alternatives to ChIP such as CUT&RUN).

We have added additional analysis on seven other *in vitro* dCas9 datasets, and have calculated both Spearman and Pearson correlation coefficients between off-target G content and CasKAS RPM values for all 12 available datasets. We observe a mixture of slightly positive and slightly negative correlations, but we do not observe strong correlations between G content and CasKAS signal. Thus our current view is that a single unpaired G is sufficient for capture of an off-target site (obviously if there is no G, that site is invisible, and that is a fundamental and unavoidable limitation of the method).

#2

This sentence should be rephrased: “It measures strand invasion by CRISPR, which is biochemically more specific and relevant to CRISPR function than DNA association.”

It is unclear that this statement is universal. For instance, certain applications of CRISPR do not require helix opening (ie Epigenetic modulation)

We have revised the sentence to better clarify what we mean. Invasion is certainly critical for cleavage, and it is also likely that transient association without strand invasion is not sufficiently stable to initiate consistent epigenetic modulation.

#3

DNA replication generates transient ssDNA exposure, also other regions such as centromeres and telomeres expose ssDNA. It would be positive to comment if this interferes with the measurement.

DNA replication indeed generates transient ssDNA exposure, but this is not a concern *in vitro*, which is the main setting in which off-targets are to be measured. *In vivo* what one is measuring is the association of Cas9 with DNA, and then such contextual features are directly evaluated by comparing dCas9/Cas9 + sgRNA profiles versus those from cells with dCas9/Cas9 but no guide.

In addition, ssDNA generated from replication, with some exceptions, does not have the same tight positioning properties as dCas9/Cas9 occupancy, which is very precisely located at the on/off-target sgRNA site. Also, the *in vivo* assay is performed on a population of cells that are unlikely to have synchronized replication fork sites.

We have revised the manuscript to make a note of replication-generated ssDNA.

Minor comments

#1 text in both figures should be made bigger. It results very hard to read the text.

We have increased font size wherever possible in the main figures.

Second round of review

Reviewer 2

I have no further comments.